# Comparison of mean platelet volume (MPV) and red blood cell distribution width (RDW) between psoriasis patients and controls: A systematic review and meta-analysis

Ping Yi[1,2☯], Jiao Jiang[1,2☯], Zheyu Wang[3☯], Xing Wang[4], Mingming Zhao[1,2], Haijing Wu[1,2]*, Yan Ding[5]*

1 Department of Dermatology, Second Xiangya Hospital, Central South University, Changsha, Hunan, China, 2 Hunan Key Laboratory of Medical Epigenomics, Second Xiangya Hospital, Central South University, Changsha, Hunan, China, 3 Department of Dermatology, Hainan General Hospital, Haikou, Hainan, China, 4 Department of Neurosurgery, West China Hospital, Sichuan University, Chengdu, Sichuan, China, 5 Department of Dermatology, Hainan Provincial Hospital of Skin Disease, Haikou, Hainan, China

☯ These authors contributed equally to this work.
* chriswu1010@126.com (HW); annymilk@126.com (YD)

**Data Availability Statement:** All relevant data are within the paper and its Supporting Information files.

## Abstract

### Background

The predictive role of hematological indexes of mean platelet volume (MPV) and red cell distribution width (RDW) has been demonstrated in cardiovascular disease concomitant with psoriasis. This meta-analysis is intended to assess whether MPV and RDW can also serve as biomarkers for the early diagnosis and disease severity assessment of psoriasis.

### Material and methods

13 studies which enrolled 1331 psoriasis patients and 919 healthy volunteers were included after screening the search results from PubMed, Embase and the Cochrane Library since inception to Mar 14, 2020. MPV of psoriasis participants and their counterparts was assessed in 10 studies, and RDW was evaluated in 4 studies, while the disease severity was measured by the Psoriasis Area and Severity Index (PASI) in 11 studies. Random-effect model analysis was applied to calculate pooled standard mean difference (SMD) with 95% confidence interval (95% CI).

### Results

Associations of MPV and RDW with the presence of psoriasis were demonstrated (MPV: SMD = 0.503, 95% CI: 0.242–0.765; RDW: SMD = 0.522, 95% CI: 0.228–0.817), but no statistically significant correlation of MPV and disease severity of psoriasis was found in meta-regression analysis (p = 0.208). Subgroup analysis revealed that the diagnosis value of MPV and RDW was consistent regardless of PASI and study type. Heterogeneity analysis between studies was implemented by chi-squared test and $I^2$ statistics. Begg's and Egger's test were utilized for the evaluation of publication bias. The sensitivity analysis revealed no significant alteration no matter which study was excluded.

**Funding:** This work was supported by the Hunan Talent Young Investigator (No. 2019RS2012), Hunan Outstanding Young Investigator (No. 2020JJ2055), National Natural Science Foundation of China (No. 81960565, No. 81560275) , Hainan Province Clinical Medical Center, Innovation Research Team Project of Natural Science Foundation of Hainan Province (No. 2018CXTD350) and the Key R&D Projects in Hainan Province (No. ZDYF2020147).

**Competing interests:** The authors have declared that no competing interests exist.

## Conclusion

MPV and RDW could serve as promising predictive diagnostic biomarkers of psoriasis.

## Introduction

Psoriasis, a common chronic and systemic autoimmune inflammatory disease, is manifested with high individual heterogeneity symptoms, including recurrent scaly and burning skin patches, thick and dented nails, as well as swollen and stiff joints, complicated with a variety of other symptoms from cardiovascular, endocrine, digestive system and mental health [1]. The diagnosis of psoriasis primarily depends on three key items—the clinical characteristics of involving skin, nails and joints that are different from differential diagnosis; frequency of disease development; the skin biopsy [2]. Given the limitation of invasiveness and hysteresis in existing diagnostic methods [3], diagnostic biomarkers can benefit patients greatly. Up to now, promising biomarker profiles of candidate gene expression (IL36G, CCL27, NOS2, C10orf99, and S100A9) [4, 5], plasma proteins (desmoplakin, cytokeratin 17, polymeric immunoglobulin receptor, and complement C3, PI3, CCL22, IL-12B) [6, 7], inflammatory cytokines (adiponectin) [8], circulating microRNAs [9], GlycA [10] have been reported to assist the clinical diagnosis and reflect disease severity and inflammation state of psoriasis. However, examinations of these diagnostic biomarkers are time-consuming and costly compared with blood routine. Gaining more insight into the diagnostic information of hematological parameters is therefore considerable.

The hematological parameters used most frequently are red blood cell (RBC), hemoglobin (Hb), hematocrit (HCT), mean corpuscular volume (MCV), red cell distribution width (RDW), white blood cell (WBC), mean platelet volume (MPV), etc. To date, several studies have confirmed the role of some hematological parameters in certain diseases. As an example, the potential value of neutrophil-lymphocyte ratio (NLR) and platelet-lymphocyte ratio (PLR) has been verified in the clinical diagnosis of ankylosing spongdylitis (AS), Behçet's disease, rheumatoid arthritis (RA) and psoriasis [11, 12]. Interestingly, MPV and RDW have been associated with the incident of primary adverse cardiac events in psoriasis vulgaris (PsV) and psoriatic arthritis (PsA) [13], and linked to inflammatory status and clinical progression in psoriasis patients. However, conflicting results have been described in an Egyptian case-control study that reported no significant difference in MPV value of psoriasis patients (n = 25) versus healthy controls (n = 25) [14]. To integrate these controversies, we screened all the published case-control studies, cohort studies and randomized control tests (RCTs) relating to this topic and conducted a meta-analysis, with the aim to uncover the association of MPV and RDW with diagnosis and disease severity of psoriasis.

## Material and methods

### Search strategy and literature screening

Two investigators searched and screened literature independently in the database of PubMed, Cochrane Library and Embase with cutoff date of Mar 14, 2020. This study was carried out following the Preferred Reporting Items for Systematic reviews and Meta-Analyses (PRISMA) statement with registration number of CRD42020178415 in the international prospective register of systematic reviews PROSPERO. Details of searching and screening were shown in S1 Table.

## Inclusion and exclusion standard

The criteria of literature inclusion and exclusion were formulated and inspected by all authors and executed by two independent authors. A protocol agreement was reached by all authors. The inclusion and exclusion standards are listed as follows:

### Inclusion standards

1. case-control studies, cohort studies and RCTs measuring the value of MPV and/or RDW in psoriasis group with or without PsA and healthy control group;

2. patients were not treated with systemic drugs or prescribed with any drugs affecting hematological indexes including antiplatelet drugs and nonsteroid anti-inflammatory drugs at least 2 weeks before the initiation of the project;

3. published in English.

### Exclusion standards

1. the review, case report, conference abstract, letters, meta-analysis, sequencing data, bioinformatics analysis, and retracted articles;

2. either the object of studies were not psoriasis patients or only male/female psoriasis patients;

3. patients with cardiac diseases, hypertension, diabetes mellitus, obesity, dyslipidemia, pregnancy, hematological diseases and any other diseases interfering the hematological parameters;

4. unavailable access and misleading data.

## Data extraction and quality assessment

The data were extracted from 13 studies by two investigators independently, inclusive of first-author name, publication year, country, study type (prospective/retrospective study), sample size, gender, MPV, RDW, the Psoriasis Area and Severity Index (PASI) and hematology analyzer. SMDs and 95% CIs were calculated using random-effect model. The literature quality was assessed with Newcastle-Ottawa Scale (NOS), and NOS score of more than 6 indicates the adequate quality of the research.

## Statistical analysis

Review Manager (RevMan 5.3) and Stata (Version 14.0) were used to analyze all extracted data. The associations of MPV and RDW with psoriasis were measured by pooling the SMD and 95% CI of each individual study. Heterogeneity analysis between studies was implemented by chi-squared test and $I^2$ statistics. Random-effect model is suitable for the case of significant heterogeneity ($I^2 > 50\%$, $p < 0.05$), unless this case, fixed-effect model is appropriate. Potential sources of heterogeneity should be interrogated by subgroup, sensitivity and meta-regression analysis in the condition of high heterogeneity. Begg's and Egger's test were used to detect publication bias.

## Results

### Included literature

The flowchart of screening and selecting process was shown in Fig 1. A total of 134 studies were identified initially and then 30 duplicated articles were removed. After reviewing the article title and abstract, 84 researches were further excluded by following inclusion and exclusion criteria. Further assessment of full-text excluded seven of the remaining 20 studies, of which two studies were excluded as patients were concomitant with metabolic syndrome that could interfere hematological indexes. Finally, 13 qualified studies were included for meta-analysis.

### Characteristics of the included studies

The major characteristics of 13 studies [14–26] enrolling 1331 psoriasis patients and 919 healthy controls were shown in Table 1. Twelve studies from Asia region, and one from Egypt, Africa were included. As to the type of study design, eleven studies were prospective and two were retrospective. Nine studies reported MPV, three reported RDW and one reported both. Among them, one study by D.S. Kim, et al. [18] carried out two parallel case-control studies in psoriasis patients with high (PASI≥10, n = 52) or low PASI (PASI<10, n = 124) and healthy controls (n = 101). All included studies were scored more than 6 by NOS and defined as high quality (median,7.0 points; range, 6–8).

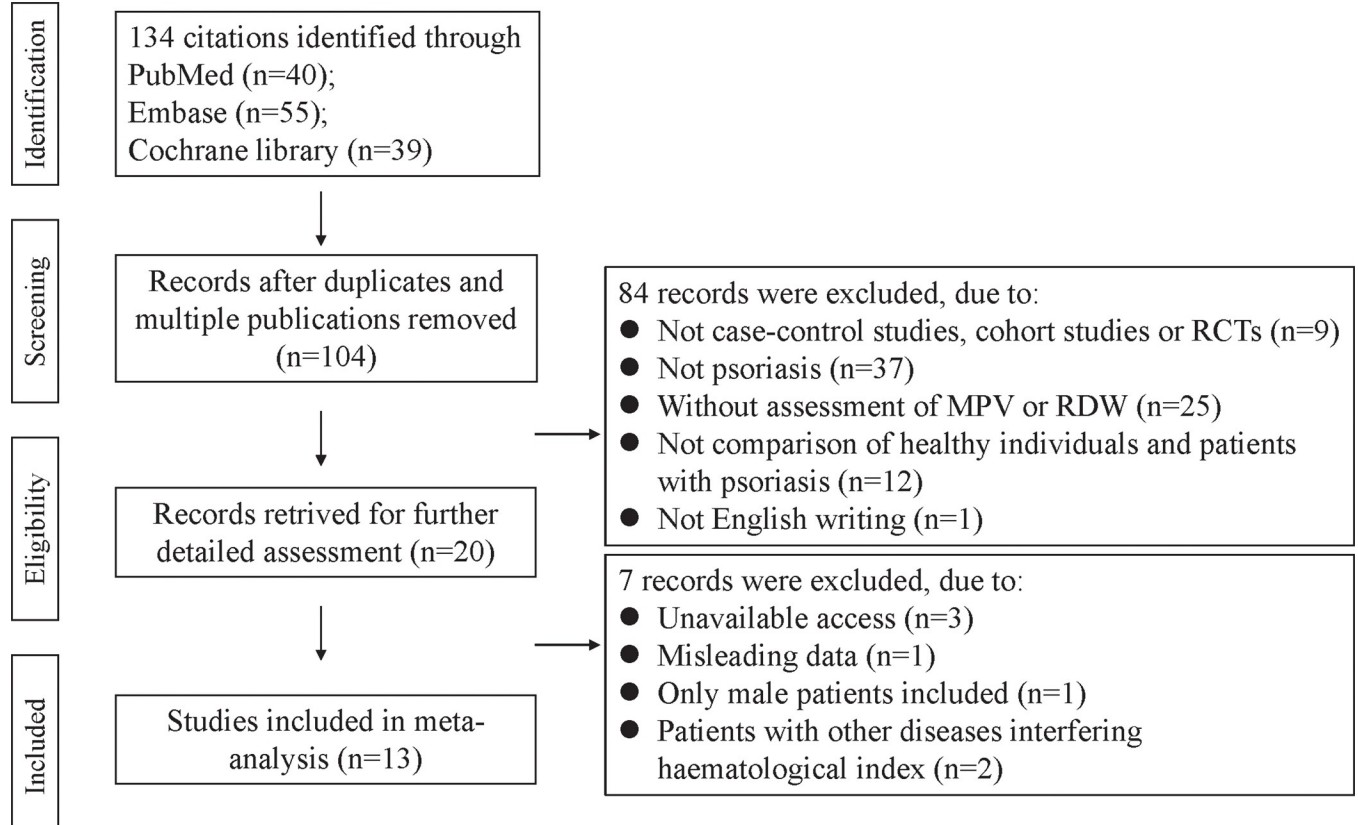

**Fig 1. Flowchart of literature screening in the meta-analysis.**

**Table 1. General characteristic of the included studies.**

| First Author | Year | Country | Study type | NOS | Psoriasis | | | | | | Control | | | | | Hematology analyzer |
|---|---|---|---|---|---|---|---|---|---|---|---|---|---|---|---|---|
| | | | | | N | Gender (M/F) | Age (Mean ± SD) | MPV (Mean ± SD) | RDW (Mean ± SD) | PASI (Mean ± SD) | N | Gender (M/F) | Age (Mean ± SD) | MPV (Mean ± SD) | RDW (Mean ± SD) | |
| F. Canpolat | 2010 | Turkey | P | 7 | 106 | 59/47 | 41.4±14 | 8.7±1.1 | NA | 13.6±6.4 | 95 | 44/51 | 40.6±8 | 7.3±0.8 | NA | NA |
| H.M.A. Saleh | 2013 | Egypt | P | 7 | 25 | 13/12 | 31.56±10.09 | 9.16±1.28 | NA | 22.59±18.07 | 25 | 12/13 | 26.52±8.73 | 9.96±1.85 | NA | Coulter LH 750 Analyzer |
| S. Koç[a] | 2013 | Turkey | P | 6 | 57 | NA | 41.8±10.8 | 8.56±0.90 | NA | 7.8±7.4 | 60 | NA | 40.0±9.4 | 8.19±0.74 | NA | NA |
| S. Koç[b] | 2013 | Turkey | P | 6 | 51 | NA | 42.1±11.1 | NA | 14.1±1.6 | 7.8±7.4 | 55 | NA | 40.1±13.1 | NA | 13.4±1 | NA |
| Z. Ahmad | 2014 | Pakistan | P | 7 | 30 | 16/14 | 40.23±10.40 | 8.24±1.22 | NA | ≥10 | 30 | 17/13 | 35.20±9.73 | 7.29±0.77 | NA | Medonic-M seires Hematology Analyzer |
| D.S. Kim[a] (1) | 2015 | Korea | R | 7 | 124 | 79/45 | 40±15.63 | 9.862±0.75 | NA | <10 | 101 | 42/59 | 37.02±15.44 | 9.724±0.59 | NA | Sysmex XE-2100 |
| D.S. Kim[a] (2) | 2015 | Korea | R | 7 | 52 | 36/16 | 39.40±14.12 | 10.08±0.67 | NA | ≥10 | 101 | 42/59 | 37.02±15.44 | 9.724±0.59 | NA | Sysmex XE-2100 |
| D.S. Kim[b] | 2015 | Korea | R | 7 | 261 | 160/161 | 39.40±15.9 | NA | 13.6±3.5 | NA | 102 | 47/55 | 36.1±13.8 | NA | 12.9±0.7 | Advia 2120 Hematology Analyzer |
| M. Unal | 2016 | Turkey | P | 6 | 320 | 154/166 | 37.34±13.45 | 8.248±1.15 | NA | NA | 200 | 111/89 | 35.89±8.65 | 7.442±1.626 | NA | NA |
| V. Raghavan | 2017 | India | P | 8 | 50 | 38/12 | 46.10±11.99 | 9.65±2.07 | 15.16±3.88 | 15.88±2.51 | 50 | 41/7 | 50.62±14.78 | 8.51±1.64 | 13.66±1.21 | Coulter LH780 Hematology Analyzer |
| Azza G. A. Farag | 2018 | Turkey | P | 7 | 70 | 40/30 | 41.47±8.16 | 9.65±1.12 | NA | 12.85±4.97 | 60 | 40/20 | 40.0±10.30 | 8.92±0.78 | NA | Sysmex XN 1000 cell counter |
| Selma Korkmaz | 2018 | Turkey | R | 7 | 38 | 22/16 | 43±9.3 | 10.2±0.9 | NA | 2.78±2.05 | 35 | 18/17 | 42±10.4 | 10±1.7 | NA | Sysmex XE-2100 |
| S.D. Pektas | 2018 | Turkey | P | 7 | 87 | 53/34 | 34.5±10.4 | NA | 13.625±0.837 | 10.2±3.7 | 76 | 42/34 | 32.7±9.7 | NA | 13.0±0.583 | Advia 2120 Hematology Analyzer |
| G.O. Yavuz | 2019 | Turkey | P | 6 | 60 | 30/30 | 40.67±15.96 | 8.9±0.96 | NA | 8.1±5.2 | 30 | 15/15 | 41.67±11.68 | 8.8±0.89 | NA | NA |

NOS: Newcastle–Ottawa quality assessment scale for case–control studies; MPV: mean platelet volume; RDW: red cell distribution width; PASI: Psoriasis Area and Severity Index; P: prospective study; R: retrospective study; NA: not available.

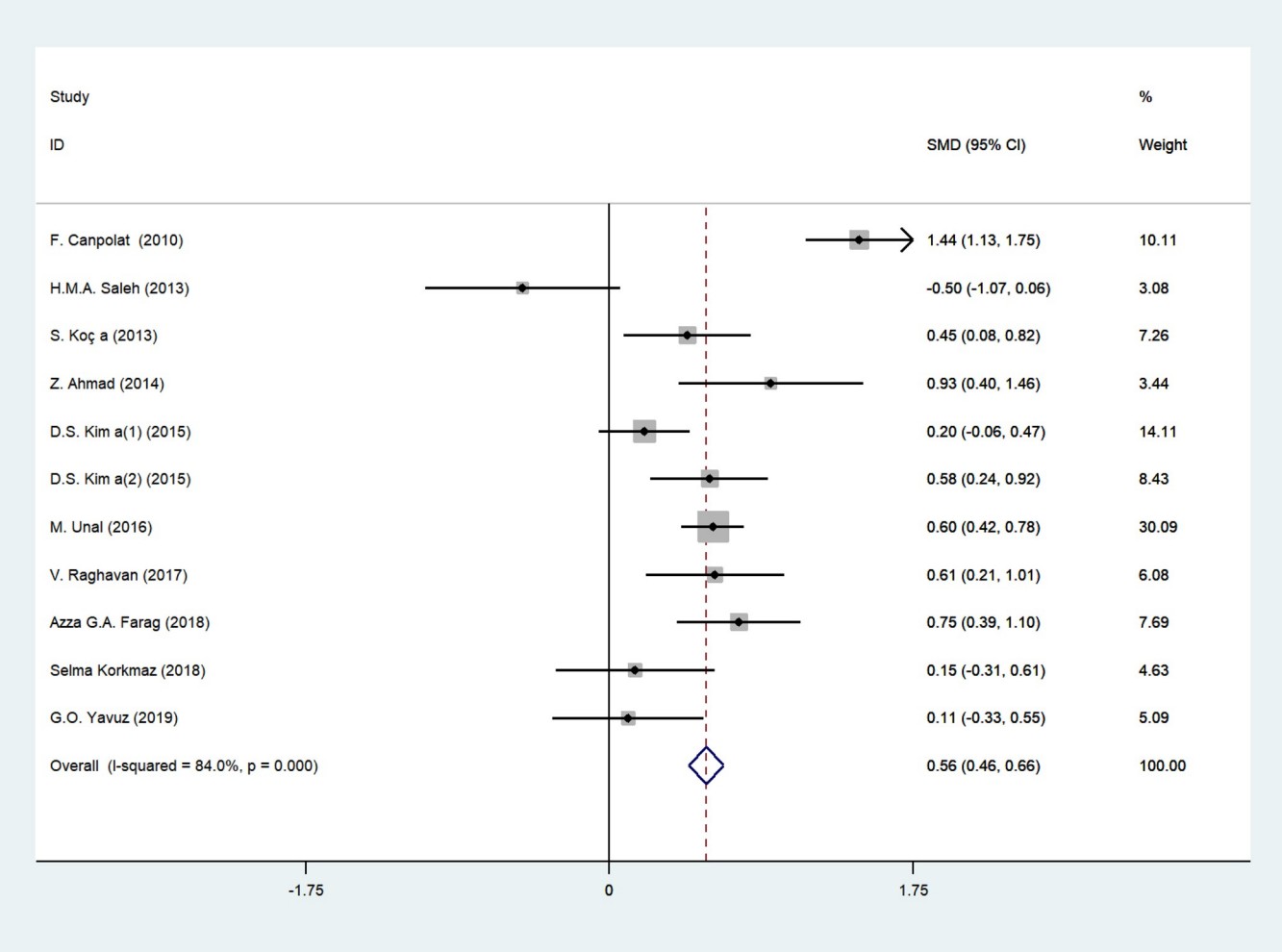

**Fig 2. Forest plot of MPV values and the presence of psoriasis.**

## MPV and psoriasis

The forest plot for MPV of 932 psoriasis patients and 686 age-matched healthy controls in 10 studies was shown in Fig 2. All the participants were adults except for one study by M. Unal, et al. [24] that enrolled both adults and children. High variation of PASI values indicated high heterogeneity in disease severity of included patients. MPV values of psoriasis patients were elevated compared with healthy subjects in 9 studies, however, one study by H.M.A. Saleh [14] showed the opposite correlation of MPV and psoriasis with no statistical significance (P = 0.085). In this case, random-effect model analysis was used ascribing to substantial heterogeneity among studies ($I^2$ = 84%, p = 0.000). Pooled SMDs revealed that MPV values were remarkably upregulated in psoriasis patients compared with healthy counterparts (SMD = 0.503, 95% CI: 0.242–0.765; p = 0.000).

The results of this analysis were credible because of the fixed pooled SMDs after removal of any individual study in sensitivity analysis, with the effect size ranging from 0.412 to 0.585 (Fig 3A). There was no significant publication bias manifested by Egger's test (P = 0.511) (Fig 4A) and Begg's test (P = 0.350) (Fig 4B).

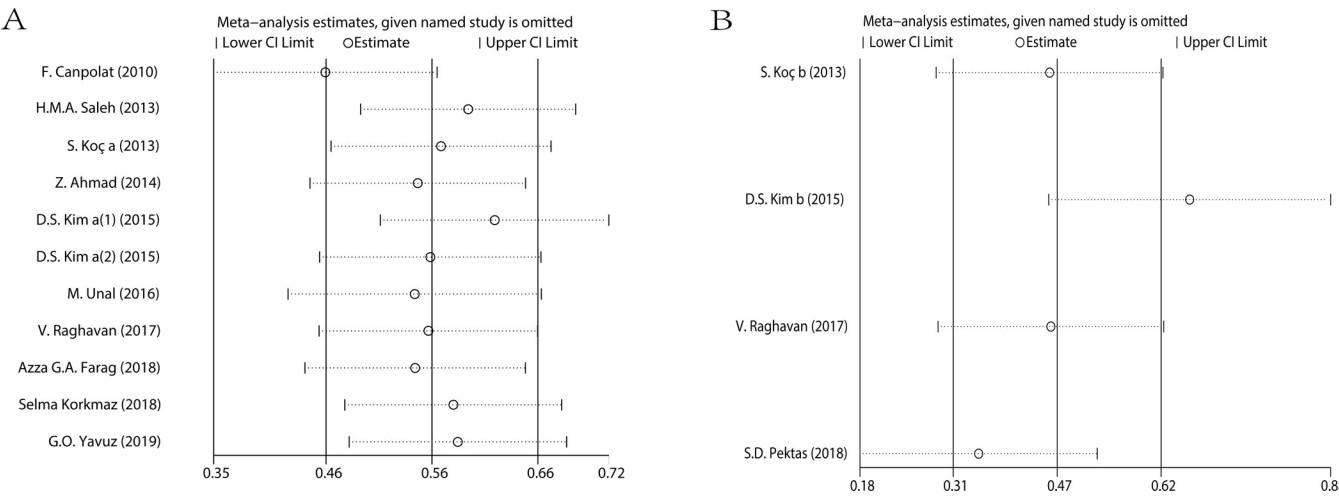

**Fig 3.** Sensitivity analysis of the association of MPV (A), RDW (B) and the presence of psoriasis.

**Fig 4.** Publication bias of studies assessing MPV and RDW: Egger's and Begg's test of MPV (A and B) and RDW (C and D).

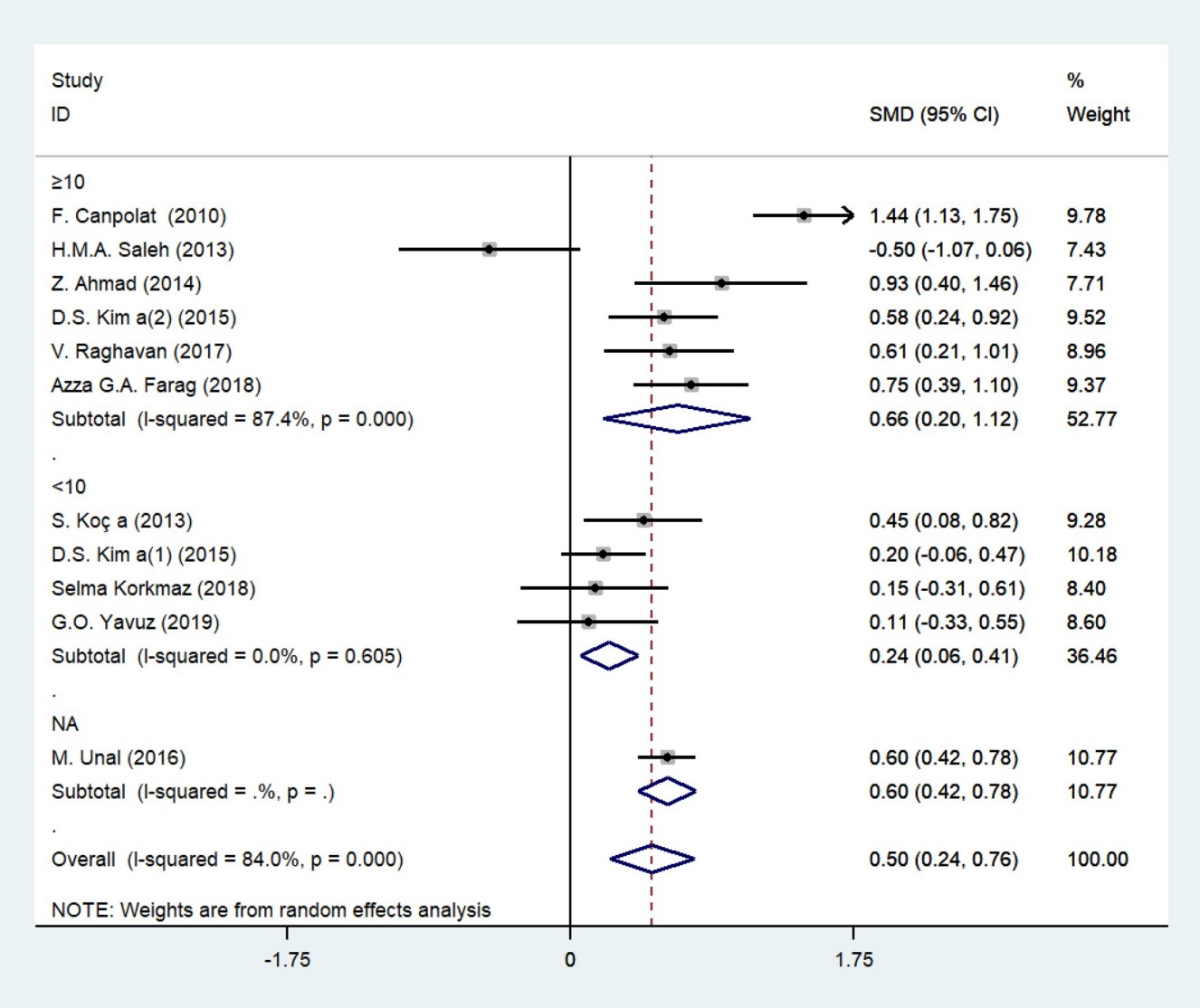

**Fig 5. Stratified analyses of disease severity assessed by PASI for the association between MPV and psoriasis.**

We applied subgroup analysis to investigate the potential variances affecting the pooled SMDs, containing PASI (PASI < 10 or PASI≥10, equal to mild psoriasis or moderate to severe psoriasis) and study type (retrospective or prospective study) (Figs 5 and 6, Table 2). SMD of MPV in patients with moderate to severe psoriasis (SMD = 0.659, 95% CI 0.199–1.118, $p = 0.005$; $I^2 = 87.4\%$, $p = 0.283$) was higher than that of patients with mild psoriasis (SMD = 0.236, 95% CI 0.059–0.414, $p = 0.009$; $I^2 = 0.0\%$, $p = 0.605$), while MPV was lower in retrospective studies (SMD = 0.315, 95% CI 0.054–0.576, $p = 0.018$; $I^2 = 42.1\%$, $p = 0.178$) compared with prospective ones (SMD = 0.573, 95% CI 0.237–0.909, $p = 0.001$; $I^2 = 86.0\%$, $p = 0.000$). However, the statistical differences were not identified in meta-regression analysis, with $P$ value of 0.208 and 0.459, respectively.

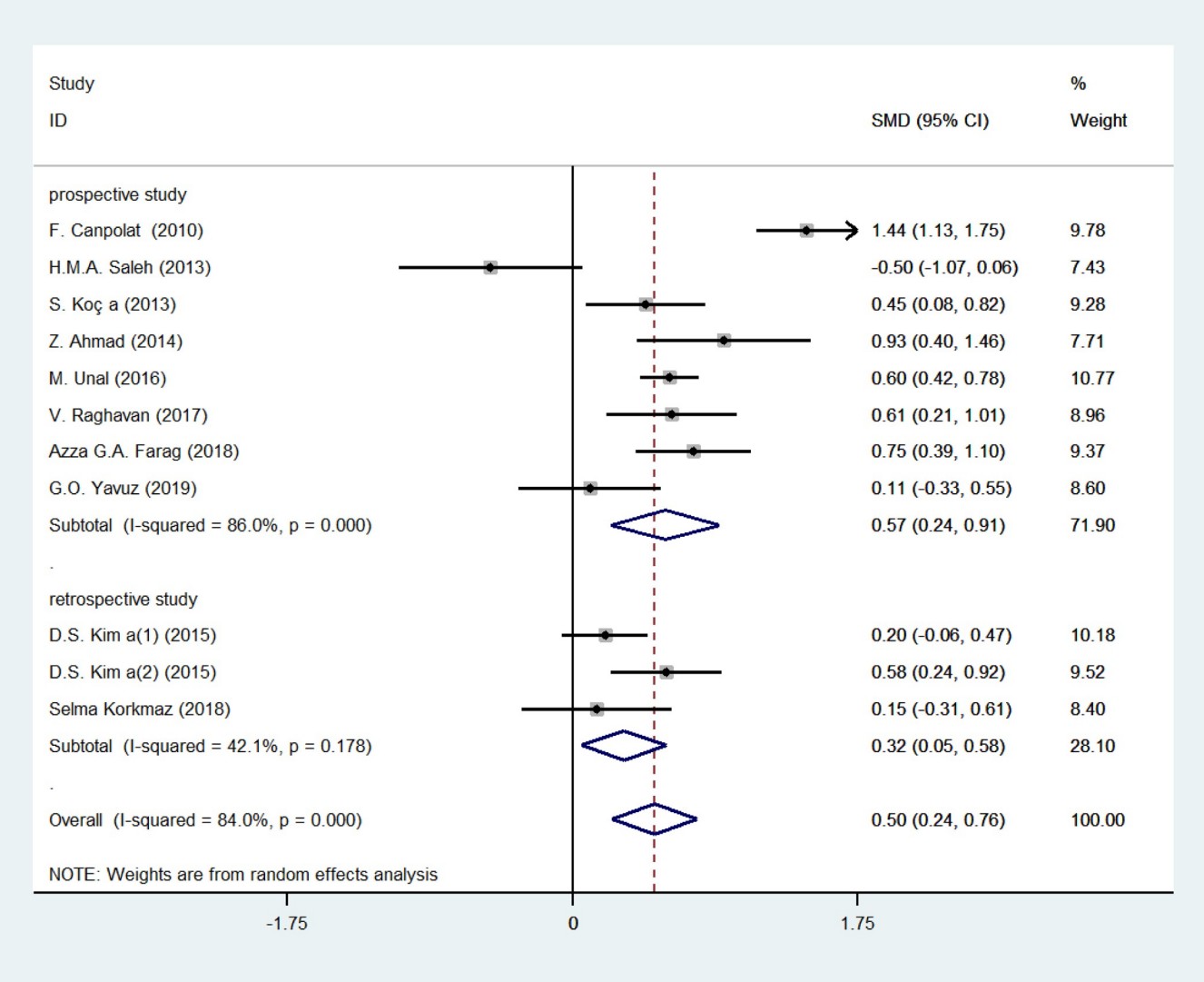

**Fig 6. Stratified analyses of study type for the association between MPV and psoriasis.**

**Table 2. Subgroup analysis of MPV in psoriasis.**

| Stratification group | N | SMD (95%CI) | Heterogeneity test | |
|---|---|---|---|---|
| | | | P | $I^2$(%) |
| Total | 11 | 0.503(0.242,0.765) | 0 | 84% |
| PASI | | | | |
| < 10 | 4 | 0.236(0.059,0.414) | 0.605 | 0 |
| ≥10 | 6 | 0.659(0.199,1.118) | 0.283 | 87.4 |
| study type | | | | |
| prospective study | 8 | 0.573(0.237,0.909) | 0 | 86 |
| retrospective study | 3 | 0.315(0.054,0.576) | 0.178 | 42.1 |

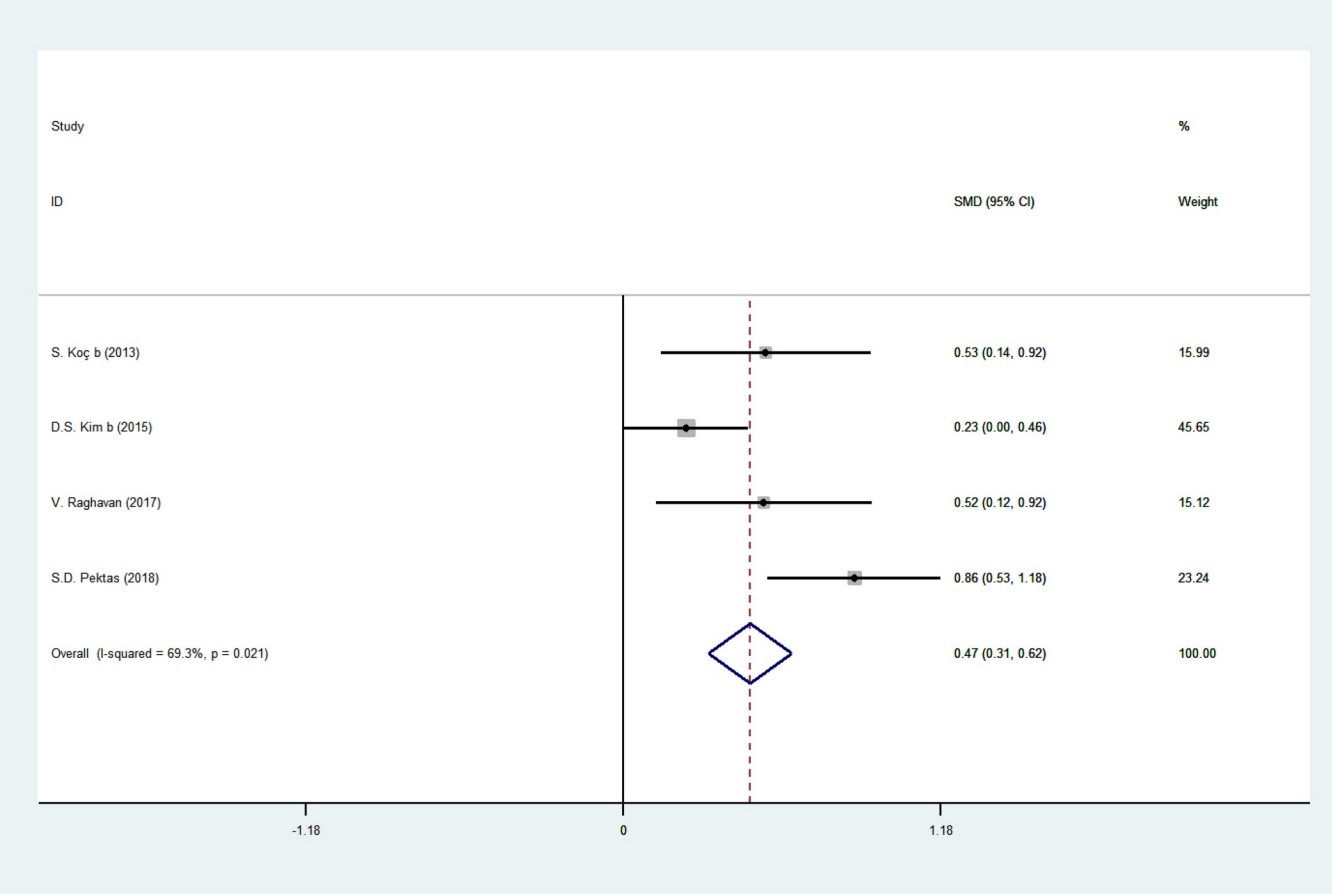

**Fig 7. Forest plot of RDW values and the presence of psoriasis.**

### RDW and psoriasis

The forest plot for RDW of 449 psoriasis patients and 283 age-matched healthy controls in four studies was shown in Fig 7. RDW values in psoriasis patients were uncovered higher than healthy matching in four studies, and pooled SMDs also showed elevated RDW values in psoriasis patients (SMD = 0.522, 95% CI: 0.228–0.817, $p$ = 0.001). Likewise, random-effect model analysis was applied due to substantial heterogeneity ($I^2$ = 69.3%, $p$ = 0.021).

The pooled SMDs were fixed in sensitivity analysis, with the effect size ranging from 0.371 to 0.662, confirming the consistency of the results of this analysis (Fig 3B). No significant publication bias was evidenced by Egger's test (P = 0.367) (Fig 4C), as well as Begg's test (P = 1.000) (Fig 4D).

## Discussion

MPV refers to the average size of circulatory platelets derived from megakaryocytes in bone marrow, functioning as the key mediators of hemostasis and thrombosis, and thus a sensitive indicator of platelet activation or reactivity [27]. Nowadays, accumulating evidence has revealed the crucial role of platelets in immune-related and inflammatory diseases [28]. Mechanically, they store multiple immune-related chemokines, cytokines, and adhesion receptors in granules and synthesize thromboxane A2, interleukin (IL)-1, and platelet-activating factor (PAF) [28]. Once activated, platelets can express or upregulate P-selectin, CD40 ligand,

Toll-like receptors, and integrins on their surface and release soluble agents such as chemokines, cytokines, 5-hydroxytryptamine (5-HT) and antimicrobial peptides to interact with immune cells [28]. Psoriasis is a chronic inflammatory Th17-driven skin disease with hyperkeratosis, dilated dermal blood vessels as well as infiltration of T cells and neutrophils [29]. Interestingly, platelet activation was found in the skin lesions of psoriasis instead of normal skin tissues and correlated with the disease severity [30].

To date, platelets were found largely involved in the pathogenesis of psoriasis. Firstly, enhanced polymorphonuclear neutrophils (PMNs) infiltrations in the psoriatic lesion and blood were in relation to the platelet surface antigens as well as the recruitment and inflammatory response caused by the soluble mediators [31]. Treatment depleting circulating platelets in imiquimod (IMQ)-induced psoriasis-like mouse model could significantly ameliorate disease severity and reduce the PMNs or platelet infiltration [31]. Moreover, many chemokines, cytokines and specific markers released or upregulated by platelets might trigger immune process. P-selectin was highly expressed on the surface of platelet in psoriasis, and could increase the aggregation of platelets and leukocytes as well as leukocyte rolling in murine skin [32]. Platelet factor 4 (PF4) and β-thromboglobulin (β-TG), the most abundant CXC chemokines of platelet α-granules, were found increased in human psoriasis and associated with PASI scores [33]. They were related to the adhesion and migration of neutrophil granulocytes and monocytes [34]. Besides, PF4 could increase IL-17 producing cells in $CD4^+$ T cells [35]. 5-HT was another molecule expressed in psoriatic lesions, leading to the immune infiltration by increasing vascular permeability, and inhibition therapy of 5-HT was effective in patients with psoriasis [36–38]. The cytokine IL-1β was also involved in the skin inflammation of psoriasis via IL-1β-IL-1R Signaling Pathway [39].

Despite the close relation between platelets and psoriasis, controversies on diagnostic value of MPV in current studies are unignorable. For instance, opposite correlation between MPV and the presence of psoriasis has been identified in an Egyptian study by H.M.A. Saleh, et al. [14], while others indicated a positive association. A meta-analysis is therefore indispensable to address this controversy. In general, our meta-analysis has verified the positive correlation by pooling results from all included studies. Meanwhile, the meta-regression analysis of PASI showed the upward trend of MPV with disease severity, though the conclusion was not statistically significant. The moderate to severe psoriasis subgroup showed significant heterogeneity due to a large span of PASI of different participants with high clinical heterogeneity. Overall, our findings integrated scattered studies and further revealed the association of MPV and the presence of psoriasis rather than disease severity. Interestingly, one published meta-analysis assessing the MPV value and Systemic Lupus Erythematosus Disease Activity Index (SLEDAI) score of SLE patients observed no relationship between MPV and disease activity of SLE, although the platelet activation plays a certain role in the pathogenesis of SLE [40].

Notably, it has been reported that MPV decreased with the variation degree associated with PASI after two-year systematic treatment in a small-sample retrospective study (n = 12) [17], but increased after 12-month infliximab or adalimumab treatment in a Japanese retrospective study enrolling patients with PsV (n = 186) and PsA (n = 50) [41]. The change of MPV value before and after treatment involves with a large spectrum of factors such as therapy type and duration, treatment respond and inflammation status of individuals, and more related clinical prospective researches are needed to reveal the correlation of MPV value in pre-/post-treatment and disease status.

RDW measures the variation of volume and size of RBCs that take part in inflammatory processes and coagulation. Inflammation disturbs erythroid maturation via membrane-bound β-receptor glycoprotein 130 (gp130) to elevate the value of RDW [42]. Inflammation-induced hydroxyl radicals and cytokines influence erythropoiesis [43]. It was reported to be positively

associated with serum inflammatory indexes—high-sensitivity C-reactive protein (hsCRP) and erythrocyte sedimentation rate (ESR) in many diseases [44]. Higher RDW was found in patients with Alzheimer's disease (AD), and it was considered as a marker of inflammation, AD presence and AD severity reflected by cognition status [45]. RDW also serves as a biomarker of cardiovascular disease susceptibility and inflammation levels in RA and AS [46]. The positive correlation of RDW with pro-inflammatory cytokines TNF-α and IL-6, as well as the negative association with anti-inflammatory cytokine IL-10 have proven the function as an auxiliary inflammatory biomarker in RA [47]. Upregulated inflammatory cytokines in the skin and peripheral blood of psoriasis patients, such as interferons, TNF-a, IL-1, IL-6 and IL-10, had an effect on erythropoiesis, leading to the enhanced RDW [48]. Thus, it is generally accepted that RDW is a potential inflammatory biomarker for psoriasis, but the specific roles of RBCs in psoriasis remain unclear. One large-scale retrospective study argued that RDW was higher in psoriasis patients with no significant association with PASI and CRP [49]. Three-month standard treatment caused a temporary reduction of RDW value in moderate to severe psoriasis patients [50], suggesting the predictive role of RDW in short-term treatment respond, however, more hematological parameters like NLR and PLR should be integrated into consideration in the long-term observation [41].

Although this meta-analysis has interrogated the significant relationship between two hematological parameters and psoriasis, there are some inevitable limitations. Firstly, hematology analyzer and reference range of MPV and RDW were not uniform between studies, which led to bias from inevitable inaccuracy of measurement. Secondly, standard treatment of psoriasis other than medicine affecting platelet function would influence the value of MPV and RDW, which might interfere the analysis results. Thirdly, the results may not be applied for all races and regions because of the majority of included studies are from Asian countries including Turkey, Korea, Pakistan and India. Finally, the sample size of retrospective study accounts for a large proportion, which reduces the evidential effectiveness of the analysis, and it's even possible that the included studies are not sufficient to detect if there is a difference, thus well-designed clinical prospective trials and large scale RCTs are urgently needed to give potent evidence of our results.

## Conclusion

It has been demonstrated that the values of MPV and RDW bear closely on the presence of psoriasis, though no relationship was found to the disease severity. These two promising indicators may benefit patients by providing auxiliary information for clinicians in the early diagnosis of psoriasis. On this setting, the association merits further verification in future study.

## Supporting information

**S1 Checklist.**
(DOC)

**S1 Table. Literature searching strategy.**
(PDF)

## Acknowledgments

JJ, PY and ZYW conceived and designed the work. Material preparation, data collection and analysis were performed by PY, JJ, ZYW, XW, and MMZ. The first draft of the manuscript was written by PY, JJ and ZYW and all authors commented on previous versions of the

manuscript. PY, HJW and YD revised the manuscript, and HJW and YD contributed to article drafting, critical revision and final approval of the version to be published. All authors read and approved the final manuscript.

## Author Contributions

**Data curation:** Mingming Zhao.

**Formal analysis:** Xing Wang.

**Writing – original draft:** Ping Yi, Jiao Jiang, Zheyu Wang.

**Writing – review & editing:** Haijing Wu, Yan Ding.

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
