## [Decision Letter · Decision Letter 0]

5 Oct 2021

PONE-D-21-08384Comparison of mean platelet volume (MPV) and red blood cell distribution width (RDW) between psoriasis patients and controls: a systematic review and meta-analysisPLOS ONE

Dear Dr. Wang,

Thank you for submitting your manuscript to PLOS ONE. After careful consideration, we feel that it has merit but does not fully meet PLOS ONE’s publication criteria as it currently stands. Therefore, we invite you to submit a revised version of the manuscript that addresses the points raised during the review process.

The reviewers have identified several aspects of your study design that require further clarification, and have also pointed to a number of opportunities to improve the presentation of the manuscript. Please attend carefully to each of the points they have raised when preparing your revisions.

We look forward to receiving your revised manuscript.

Kind regards,

Jamie Males

Staff Editor

PLOS ONE

Journal Requirements:

Reviewers' comments:

Reviewer's Responses to Questions

**Comments to the Author**

1. Is the manuscript technically sound, and do the data support the conclusions?

Reviewer #1: Yes

Reviewer #2: Yes

2. Has the statistical analysis been performed appropriately and rigorously? 

Reviewer #1: Yes

Reviewer #2: Yes

3. Have the authors made all data underlying the findings in their manuscript fully available?

Reviewer #1: Yes

Reviewer #2: Yes

4. Is the manuscript presented in an intelligible fashion and written in standard English?

Reviewer #1: No

Reviewer #2: Yes

5. Review Comments to the Author

Reviewer #1: This is a meta-analysis of RDW and MPV as biomarkers and predictors of disease severity by Jiang et al.

1. Which drugs affecting hematologic diseases were excluded? Was biotin also excluded?

2. Was there a reason why male or female only studies were excluded? While there are potential gender differences in values, you could also run a subset analysis based on gender. Please include reasoning.

3. In limitations, this reviewer suggests adding that it's possible that even aggregated the included studies are not sufficient to detect if there is a difference.

Minor comments:

1. Suggest adding a transition sentence between promising biomarkers and hematologic parameters to improve flow

2. There are spelling/grammar/word errors throughout the paper and highlighting several here:

Introduction- change "The diagnose the psoriasis" to "The diagnosis of psoriasis"

Exclusion standards- change retreated to retracted

Results-11 and 12 should be spelled out

Figure 1- psoriasis is misspelled

Reviewer #2: The present meta-analysis aimed at investigating the association of MPV and RDW with diagnosis and disease severity of psoriasis by extracting data from thirteen studies appropriately chosen according to certain criteria.

The manuscript provides robust statistical analysis on the extracted data to assess the association of MPV and RDW with psoriasis, the heterogeneity among studies and the entity of the publication bias, as well as sensitivity and meta-regression analysis. Nevertheless, more details are needed to clarify the biological aspects related to MPV and RDW involvement in the pathogenesis of psoriasis.

• In the Discussion Section about MPV, authors mention findings from other studies that demonstrate elevation of this parameter in psoriasis patients, suggesting the involvement of increased platelet activation in the pathogenesis of psoriasis. Authors should give a deeper explanation on how MPV can be affected by the clinical-molecular alterations underlying the disease onset and progression, outlining possible mechanisms, pathways or molecular factors related (for instance, recent articles/reviews concerning the role of angiogenesis, cell proliferation, inflammation could be employed for a better introduction of the topic).

• Previous literature suggest RDW as a biomarker of cardiovascular disease susceptibility and inflammation levels in rheumatoid arthritis (RA) and ankylosing spongdylitis (AS). In the Discussion section, authors do not explore the connection between the variation of volume and size of red blood cells (measured by RDW) and the pathological processes underlying the development of psoriasis. Therefore, authors should explain how this parameter could be involved in the pathogenesis of the disease, justifying why clinicians should consider RDW informative in psoriasis patients especially regarding its predictive role in response to treatment.

• This meta-analysis demonstrates a correlation of MPV and RDW with the presence of psoriasis rather than with disease severity. Authors should implement the conclusion suggesting how this finding could help clinicians with the disease management, selection of therapies and patient follow-up, since they purpose these parameters as promising predictive and diagnostic biomarkers of psoriasis.

• References must have the same format even if they come from different databases to allow users easier searching on the source of information mentioned in the manuscript. Therefore, authors should review the bibliography inserting some missing information for example the Digital Object Identifier (doi) and the source of the cited article in all references.

6. PLOS authors have the option to publish the peer review history of their article (what does this mean?). If published, this will include your full peer review and any attached files.

Reviewer #1: No

Reviewer #2: No

---

## [Author Response · Author response to Decision Letter 0]

21 Oct 2021

Response to Reviewers

Thanks for your thoughtful and constructive comments. We have learned a lot from these comments, and addressed all of questions accordingly. Please find the revised texts highlighted in the revised manuscript, and the point-by-point responses as follows:

Journal Requirements:

Response: We have checked the manuscript carefully and ensured that our manuscript meets PLOS ONE's style requirements, including file naming.

Response: We have changed data availability statement to “all relevant data are within the manuscript and its Supporting Information files”, since there is no additional data for this manuscript.

Response: We have ensured that abstract on the online submission and the abstract in the manuscript are identical.

Response: We have included captions for our Supporting Information files at the end of our manuscript, and updated the in-text citation to match accordingly.

Reviewers' comments:

Reviewer #1: This is a meta-analysis of RDW and MPV as biomarkers and predictors of disease severity by Jiang et al.

1. Which drugs affecting hematologic diseases were excluded? Was biotin also excluded?

Response: Thanks for your comment. We have excluded the antiplatelet drugs and nonsteroid anti-inflammatory drugs but not biotin, because the measurement of MPV and RDW is independent of biotin in blood (certain laboratory results influenced by biotin include troponin, thyroid and other hormones and vitamin D level) (https://labtestsonline.org/articles/biotin-affects-some-blood-test-results).

2. Was there a reason why male or female only studies were excluded? While there are potential gender differences in values, you could also run a subset analysis based on gender. Please include reasoning.

Response: Thanks for your important comment. Indeed, only one monosexual study was excluded based on this criterion. The reason for establishing this standard is that all enrolled studies focused on both the male and female except one study by Ozlem Karabudak in 2008. As you mentioned, subset analysis could be used to diminish the effect of gender, however, only two included studies have provided the MPV value stratified by gender. Consequently, data is insufficient to get credible results for the subset analysis of gender. In order to avoid bias caused by gender, we have excluded this study by exclusive standard. 

3. In limitations, this reviewer suggests adding that it's possible that even aggregated the included studies are not sufficient to detect if there is a difference.

Response: We have added “it's possible that even aggregated the included studies are not sufficient to detect if there is a difference” in limitations.

Minor comments:

1. Suggest adding a transition sentence between promising biomarkers and hematologic parameters to improve flow

Response: Thanks for your suggestion. We have added a transition sentence between promising biomarkers and hematologic parameters in introduction section to improve flow.

2. There are spelling/grammar/word errors throughout the paper and highlighting several here:

Introduction- change "The diagnose the psoriasis" to "The diagnosis of psoriasis"

Response: We have changed "The diagnose the psoriasis" to "The diagnosis of psoriasis"

in introduction section.

Exclusion standards- change retreated to retracted

Response: We have changed “retreated” to “retracted” in exclusion standards.

Results-11 and 12 should be spelled out

Response: We have spelled out 11 and 12 in results.

Figure 1- psoriasis is misspelled

Response: We have corrected the spelling mistake in figure 1.

Meanwhile, we have our manuscript polished by a native English speaker to make it more readable.

Reviewer #2: The present meta-analysis aimed at investigating the association of MPV and RDW with diagnosis and disease severity of psoriasis by extracting data from thirteen studies appropriately chosen according to certain criteria.

The manuscript provides robust statistical analysis on the extracted data to assess the association of MPV and RDW with psoriasis, the heterogeneity among studies and the entity of the publication bias, as well as sensitivity and meta-regression analysis. Nevertheless, more details are needed to clarify the biological aspects related to MPV and RDW involvement in the pathogenesis of psoriasis.

1. In the Discussion Section about MPV, authors mention findings from other studies that demonstrate elevation of this parameter in psoriasis patients, suggesting the involvement of increased platelet activation in the pathogenesis of psoriasis. Authors should give a deeper explanation on how MPV can be affected by the clinical-molecular alterations underlying the disease onset and progression, outlining possible mechanisms, pathways or molecular factors related (for instance, recent articles/reviews concerning the role of angiogenesis, cell proliferation, inflammation could be employed for a better introduction of the topic).

Response: Thanks for your constructive comment! We have cited more related literation to elaborate MPV-involved possible mechanisms underlying the disease onset and progression. Platelets synthesize, store, and secret inflammatory factors to induce inflammatory response. At the same time, they express P-selectin, CD40 ligand, Toll-like receptors, and integrins on their surface for the recruitment of polymorphonuclear neutrophils, cell adhesion and migration, as well as interaction with immune cells in the onset and progression of psoriasis. 

2. Previous literature suggested RDW as a biomarker of cardiovascular disease susceptibility and inflammation levels in rheumatoid arthritis (RA) and ankylosing spongdylitis (AS). In the Discussion section, authors do not explore the connection between the variation of volume and size of red blood cells (measured by RDW) and the pathological processes underlying the development of psoriasis. Therefore, authors should explain how this parameter could be involved in the pathogenesis of the disease, justifying why clinicians should consider RDW informative in psoriasis patients especially regarding its predictive role in response to treatment.

Response: Thanks for your important comment. We pretty agree with your point of view. However, the erythroid disturbance in psoriasis patients has been hardly reported. A few studies reported that increased inflammatory cytokines in the skin and peripheral blood of psoriasis patients influenced erythropoiesis and elevated the value of RDW. We have cited and discussed these articles in the discussion section (page 17-18). Moreover, further studies are needed to have a more in-depth sight. 

3. This meta-analysis demonstrates a correlation of MPV and RDW with the presence of psoriasis rather than with disease severity. Authors should implement the conclusion suggesting how this finding could help clinicians with the disease management, selection of therapies and patient follow-up, since they purpose these parameters as promising predictive and diagnostic biomarkers of psoriasis.

Response: Thanks for your comment. The current study could only demonstrate the correlation of MPV and RDW with the presence of psoriasis. Further analysis such as the disease severity and long-term prognosis is limited by unavailable data currently. Thus, we could only point out that MPV and RDW contribute to the early diagnosis of psoriasis for clinicians. As we know, patients could benefit a lot from early diagnosis and treatment. Besides, several studies have showed that the induction of these two indicators possibly provide reference significance for the judgement of treatment efficacy in the short term, however the comprehensive analysis of several hematological parameters are needed for efficacy evaluation in the long term. We have further elucidated the above opinion in the discussion section.

4. References must have the same format even if they come from different databases to allow users easier searching on the source of information mentioned in the manuscript. Therefore, authors should review the bibliography inserting some missing information for example the Digital Object Identifier (doi) and the source of the cited article in all references.

Response: We have reviewed the bibliography and inserted some missing information including doi and the source of the cited article in all references except for Ref. 26, 46 and 50 without available doi.

---

## [Decision Letter · Decision Letter 1]

14 Feb 2022

Comparison of mean platelet volume (MPV) and red blood cell distribution width (RDW) between psoriasis patients and controls: a systematic review and meta-analysis

PONE-D-21-08384R1

Dear Dr. Wang,

We’re pleased to inform you that your manuscript has been judged scientifically suitable for publication and will be formally accepted for publication once it meets all outstanding technical requirements.

Kind regards,

Muhammad Tarek Abdel Ghafar, M.D

Academic Editor

PLOS ONE

Additional Editor Comments (optional):

Reviewers' comments:

Reviewer's Responses to Questions

**Comments to the Author**

1. If the authors have adequately addressed your comments raised in a previous round of review and you feel that this manuscript is now acceptable for publication, you may indicate that here to bypass the “Comments to the Author” section, enter your conflict of interest statement in the “Confidential to Editor” section, and submit your "Accept" recommendation.

Reviewer #2: All comments have been addressed

Reviewer #3: All comments have been addressed

2. Is the manuscript technically sound, and do the data support the conclusions?

Reviewer #2: Yes

Reviewer #3: Partly

3. Has the statistical analysis been performed appropriately and rigorously? 

Reviewer #2: Yes

Reviewer #3: Yes

4. Have the authors made all data underlying the findings in their manuscript fully available?

Reviewer #2: Yes

Reviewer #3: Yes

5. Is the manuscript presented in an intelligible fashion and written in standard English?

Reviewer #2: Yes

Reviewer #3: Yes

6. Review Comments to the Author

Reviewer #2: The present version of the manuscript is considerably improved. In particular, the revised version of the manuscript provides a more detailed and clearer explanation of MPV and RDW involvement in the pathogenesis of psoriasis clarifying its related biological aspects.

Reviewer #3: The present meta-analysis aimed at investigating the association of MPV

and RDW with diagnosis and disease severity of psoriasis. The language of the manuscript is good and the structure is reasonable.

7. PLOS authors have the option to publish the peer review history of their article (what does this mean?). If published, this will include your full peer review and any attached files.

Reviewer #2: No

Reviewer #3: No

---

## [Editor Report · Acceptance letter]

17 Feb 2022

PONE-D-21-08384R1 

Comparison of mean platelet volume (MPV) and red blood cell distribution width (RDW) between psoriasis patients and controls: a systematic review and meta-analysis 

Dear Dr. Wang:

I'm pleased to inform you that your manuscript has been deemed suitable for publication in PLOS ONE. Congratulations! Your manuscript is now with our production department. 

Kind regards, 

on behalf of

Prof Muhammad Tarek Abdel Ghafar 

Academic Editor

PLOS ONE